# A New Factor “Otherism” Added to the Hedonic and Eudaimonic Motives for Activities Scale (HEMA) in Chinese Culture

**DOI:** 10.3390/bs13090746

**Published:** 2023-09-06

**Authors:** Rong Dong, Yunxi Wang, Chenguang Wei, Xiangling Hou, Kang Ju, Yiming Liang, Juzhe Xi

**Affiliations:** 1Shanghai Key Laboratory of Mental Health and Psychological Crisis Intervention, Affiliated Mental Health Center (ECNU), Positive Education China Academy (PECA), Han-Jing Institute for Studies in Classics, Juzhe Xi’s Master Workroom, Shanghai School Mental Health Service, School of Psychology and Cognitive Science, East China Normal University, Shanghai 200062, China; 2School of Education, Chongqing Normal University, Chongqing 401331, China; 3Shanghai Changning Mental Health Center, Shanghai 200335, China

**Keywords:** orientation to happiness, collectivist culture, psychometric study

## Abstract

Due to different understandings of happiness, people adopt different tendencies to act, which is called orientation to happiness (OTH). Our previous study found that OTH had two core themes, Self-focused and Other-focused in Chinese culture, which was different from OTH structures in Western culture. However, no corresponding measurement tool has been developed or revised. The Hedonic and Eudaimonic Motives for Activities Scale (HEMA) was the most commonly used measurement tool of OTH in recent years. The present study aimed to develop a Chinese version of the HEMA. A total of 1729 Chinese adults participated in this study. The exploratory factor analysis (EFA) and confirmatory factor analysis (CFA) were used to examine the underlying structure of the Chinese version of the HEMA. The results supported the 3-factor structure of the translation instrument, and the 15-item scale had good convergent and discriminant validity. The three dimensions were named Hedonism, Eudaimonism, and Otherism. Among them, Otherism is a new dimension, which means “the pursuit of the harmony of the group and achieving happiness by fulfilling their responsibilities in the group”. The revised tool was named the Hedonic, Eudaimonic, and Otheristic Motives for Activities Scale-Chinese (HEOMA-C). The results showed that the HEOMA-C has good reliability and validity. Overall, the present study provided an effective tool to assess the OTH in Chinese culture.

## 1. Introduction

The pursuit of “happiness” is the most precious ultimate life goal for individuals [1,2,3]. Most of the previous research on happiness was based on two philosophical views: hedonia and eudaimonia [4,5]. The hedonic view believes that happiness is achieved through the pursuit of pleasure and comfort [6], while the eudaimonic view emphasizes that only meaningful activities can bring people a true and lasting experience of happiness. Orientation to happiness (OTH) is what individuals pursue when they decide to do/participate in something, including values, priorities, motivations, ideals, and goals [7,8]. Assessing hedonia and eudaimonia in orientation is a great approach to assessing underlying motivations for activities rather than the surface content of activities [9].

The culture in which individuals live can lead to different types of OTH [10]. In order to better understand the OTH under the collectivist culture, our previous study explored the structure of the OTH of Chinese adults through semi-structured interviews [11]. The results showed that the OTH had two core themes, Self-focused and Other-focused. Self-focused contained two main themes, Self-hedonism and Self-eudaimonism, which were consistent with previous research on the OTH [7,8,9,12,13,14]. Self-hedonism refers to the pursuit of individual happiness and Self-eudaimonism refers to the pursuit of individual authenticity, meaning achievement and growth. It is noteworthy that we found the core theme of Other-focused, that is, taking the harmony of the group and fulfilling the responsibilities in the group as one’s pursuit. It was further divided into two main themes: Other-hedonism and Other-eudaimonism. The former refers to the pursuit of pleasure and comfort in the interaction with significant others and maintaining a good interpersonal relationship, such as the following: “I like eating and chatting with my family. I feel very relaxed.” “I love to party and meet new people.” The latter refers to the pursuit of behaviors that provide value to important relationships, broader development and growth, and working together with others, for example as follows: “My biggest motivation now is my daughter. I want to work harder to make it easier for her in the future”. “I really want to contribute to society”. Overall, these findings showed that the Other-focused dimension is an important part of the structure of OTH in Chinese culture. However, no corresponding measurement tool has been developed or revised.

The Hedonic and Eudaimonic Motives for Activities Scale (HEMA) by Huta and Ryan [7] has been the most commonly used measurement tool of OTH in recent years. Later, Huta and Ryan developed the revised version of the Hedonic and Eudaimonic Motives for Activities Scale (HEMA-R) [9]. The revised version asks participants to assess the reasons why they decided to do something (e.g., “Seeking relaxation”, “Seeking to develop a skill, learn, or gain insight into something”). Some Chinese researchers have tested the relevant indicators of the HEMA scale [15], translated the original scale, and tested it in Chinese groups. However, they did not deviate from the original framework to explore OTH in Chinese collectivist culture. In general, the current OTH measurement tool does not include the Other-focused dimension of OTH in Chinese culture, so further revision is needed.

The purpose of the present study was to conduct a Chinese revision of HEMA based on our previous study of OTH structure in Chinese culture. In addition, we test the reliability and validity of the revised HEMA.

## 2. Materials and Methods

### 2.1. Design and Participants

This research was a cross-sectional study conducted in China. A total of 1729 people participated in the survey. The participants all provided informed consent before participating in this study. The inclusion criteria were as follows: (a) aged ≥ 18 years, (b) native Mandarin speakers, and (c) without communication impairment (deaf or blind). The present study was approved by the University Committee on Human Research Protection of East China Normal University (HR2-0065-2021).

Sample 1 was used for questionnaire formation and structural exploration. We recruited participants using convenience sampling at a university in Shanghai and collected 479 participants (58.45% of them were female). The average age of the participants was 21.28 years (*SD* = 4.25), ranging from 18 to 30 years. Among 479 participants, 309 (64.6%) were born in urban areas and 170 (35.4%) in rural areas; 215 (44.9%) were only children and 264 (55.1%) were non-only children.

Sample 2 was used to assess most of the psychometric properties of the scale. We distributed the questionnaire through the online data collection platform “credamo”. A total of 511 participants completed the administration, of whom 235 (54.01%) were male; 307 (60.1%) were born in urban areas and 204 (39.9%) in rural areas; 231 (45.2%) were only children and 280 (54.8%) were non-only children. The average age of the participants was 28.91 years (*SD* = 3.13), ranging from 18 to 40 years.

Sample 3 was used to assess the retest reliability and longitudinal measurement invariance. Four follow-up tests were conducted in September (T1), October (T2), November (T3), and December (T4) in 2022. The data were obtained through the online data collection platform “credamo”. A total of 539 questionnaires were collected by T1, including 199 males (36.9%) and 340 females (63.1%), with a mean age of 24.98 years old (*SD* = 6.62), ranging from 18 to 52 years old. The following three tests all had a loss of participants, among which 520 participants completed T2, 400 participants completed T3, 427 participants completed T4, and 363 completed all four tests.

Sample 4 was used to assess the criterion validity. We distributed the questionnaire through the online data collection platform “credamo”. A total of 200 adults participated in the test, of whom 38.5% were male. The average age of the participants was 29.21 years (*SD* = 4.21), ranging from 18 to 50 years.

### 2.2. Instrumentation

#### 2.2.1. The Revised Version of the Hedonic and Eudaimonic Motives for Activities Scale (HEMA-R)

The HEMA-R was developed by Huta and Ryan [7] and subsequently revised [9]. The revised version comprises 10 items in 2 dimensions, Hedonia and Eudaimonia, with 5 items in each dimension. The scale is rated on a 7-point scale from 0 (not at all) to 7 (very much). The scale asks participants to assess the reasons why they decided to do something (e.g., “Seeking to do what you believe in?”, “Seeking pleasure?”). Higher scores indicate higher motivation on that dimension and stronger OTH. The scale has had high internal consistency in previous studies [9].

#### 2.2.2. Orientation to Happiness Scale (OHS)

The OHS developed by Peterson, Park, and Seligman [8] was used as a criterion because previous studies [16,17] showed that dimensions in the OHS were moderately correlated with the HEMA-R. The OHS has three subscales: meaning (e.g., “My life serves a higher purpose”), pleasure (e.g., “For me, a pleasurable life is a good life”), and engagement (e.g., “In choosing what to do, I always take into account whether I can lose myself in it”). Participants were given a 5-point scale for how well each description matched their own (1 = not at all, 5 = completely). The Chinese version of the scale was translated and validated by Chen [18]. In this study, the OHS had good internal consistency (α = 0.80).

#### 2.2.3. The Beliefs about Well-Being Scale (BAWB)

The BAWB was developed by McMahan and Estes [19]. It consists of 16 items divided into 4 dimensions, with each dimension containing 4 items. The dimensions are the experience of pleasure (e.g., experiencing euphoria and pleasure), avoidance of negative experiences (e.g., a lack of painful experiences), self-development (e.g., working to achieve one’s true potential), and contribution to others (e.g., living in ways that benefit others). Participants rate each item on a 7-point scale based on their own beliefs (1 = strongly disagree, 7 = strongly agree). A team of four researchers translated it into Chinese. In this study, the BAWB had good internal consistency (α= 0.76).

#### 2.2.4. Complexity of Happiness Definitions and Intentions (CoDI)

The CoDI scale was developed by Krasko, Intelisano, and Luhmann [20]. It consists of 24 items divided into 8 subscales, with each subscale containing 3 items. Each item includes two questions: happiness and well-being (HWB) definitions (e.g., for me personally, happiness means…) and HWB intentions (e.g., in daily life, I try…). The subscales include absence of negativity (e.g., not to worry), positive attitude (e.g., to go through life cheerfully), tranquility (e.g., a peaceful state of mind), personal development (e.g., to develop beyond oneself), luck (e.g., to be favored by luck or fate), joy and desires (e.g., experiencing a great deal of pleasure), purpose (e.g., to have a clear meaning in life), and belonging (e.g., to feel close to others). Participants rated each item on a 6-point scale based on their own beliefs (1 = does not apply, 6 = applies completely). A team of four researchers translated it into Chinese. In this study, CoDI had good internal consistency (α_definition_ = 0.75, α_intention_ = 0.81).

### 2.3. Procedure

#### 2.3.1. Translation of the Original Questionnaire

First, a doctoral student in psychology and an MA in translation translated the HEMA-R into Chinese. By comparing the two translations topic by topic, a first draft was produced that was consistent with the original meaning and professional expression while also conforming to Chinese language conventions. Second, another Ph.D. candidate in psychology (who had not seen the HEMA-R before) performed the back translation. Third, the back translation was compared with the original manuscript, and a small number of controversial topics were discussed and revised to produce a Chinese translation of the HEMA-R.

#### 2.3.2. Formation of the Initial Questionnaire 

First, based on the results of our previous study [11], 5 items on each of the two dimensions of Other-hedonism and Other-eudaimonism were developed. Combined with the Chinese translation of the HEMA-R to produce the original questionnaire, the initial questionnaire consisted of 20 items. Second, two psychology Ph.D. candidates rated the content of the questionnaire, tested its content validity, and commented on the readability and appropriateness of each item. They also revised the items where the semantics were unclear. Third, the translated questionnaire and the additional items were validated by a psychology professor. Fourth, 20 non-psychology undergraduates (half of each gender) were selected to take a trial test of the items, and the participants all gave the feedback that there was no ambiguity in the descriptions of the items in the questionnaire and that there was no confusion in the process of filling in the items, thus forming the 20-item “Activity Motivation Measurement Scale (Chinese Preliminary Test Version)”.

#### 2.3.3. Initial Questionnaire Testing

The Activity Motivation Measurement Scale (Chinese Preliminary Test Version) was distributed among students at a university in Shanghai. Five counselors from three faculties were first contacted to indicate the purpose and content of the study. After obtaining their consent, they were trained on standardized procedures and ethical guidelines for investigations. The teachers distributed the online questionnaire through WeChat groups in class meetings. Students completed the informed consent form before completing it. The informed consent form clearly informs participants that participation is voluntary and confidential and that they are free to withdraw in the middle of the process without any penalty. Participants gave their informed consent before formally completing the online questionnaire.

#### 2.3.4. Retesting and Longitudinal Measurement 

To test the psychometric properties, we collected sample 2 and sample 4. To test the retest reliability and the longitudinal measurement invariance, four follow-up tests were conducted in September (T1), October (T2), November (T3), and December (T4) in 2022. The questionnaire was posted through an online platform, and participants read the informed consent form and gave their consent before starting to fill it out. After completing the questionnaire, each participant received CNY 3 as a reward.

### 2.4. Data Analysis

We used IBM SPSS Statistics version 26 to perform descriptive statistics and exploratory factor analysis. Confirmatory factor analysis was performed by Mplus version 8.3. There are no missing values in the data.

## 3. Results

### 3.1. Item Analysis

Based on classic measurement theory, the scores of participants on each theoretical dimension were summed up. The lowest 27% and highest 27% of participants were selected as the low-scoring group and high-scoring group, respectively, based on the total scores of each dimension. The group difference test and item–total correlation were used as the discrimination indicators to test the discrimination of each item. After the test, the discrimination of all items (*t* = 17.22~25.71) and the item–total correlation (*r* = 0.70~0.87) were significant (*ps* < 0.001), so no items were removed in the item analysis.

### 3.2. Exploratory Factor Analysis

A preliminary examination of the measurement tool was conducted using Sample 1. The results of the KMO and Bartlett’s test indicated that KMO = 0.90 and Bartlett’s test of sphericity were significant (χ^2^ = 5412.76, *df* = 190, *p* < 0.001), indicating that the measurement tool was suitable for exploratory factor analysis.

Exploratory factor analysis was conducted on 20 items, utilizing principal component analysis with an optimal oblique rotation. Three factors were extracted with eigenvalues greater than one. Furthermore, an analysis was conducted on the factor loading of each item, revealing that five items exhibited “double loading”, whereby they loaded above 0.50 on two factors simultaneously. These double-loading items were then sequentially removed based on the degree of similarity in their scores across the two factors, with one item being removed at a time, followed by exploratory factor analysis. Ultimately, 15 items were retained, of which 10 were from the original HEMA-R and 5 were new additions.

The final 15 items were used for a second exploratory factor analysis by principal component analysis and optimal oblique rotation, KMO = 0.88. Bartlett’s test of sphericity was significant (χ^2^ = 3433.79, *df* = 105, *p* < 0.001). The results of the exploratory factor analysis are presented in Table 1.

Finally, a three-factor model was obtained. The model explains 63.78% of the total variable, and the variance explained that the rates of the three rotated factors’ eigenvalues are 3.41, 3.16, and 2.98. The factor loadings for each factor are all above 0.60, and the communalities are all above 0.40.

In addition, the discrimination index and the total correlation of each item were also investigated. The discrimination index was calculated using the 27/73 percentile method by subtracting the mean score of the high-scoring group from that of the low-scoring group for each item and then dividing it by the group range. This provided the discrimination index value for each item. The total correlation of each item was measured as the correlation between the item and its corresponding dimension. The study showed that the discrimination index value for each item was above 0.33, and the total correlation for each item was above 0.70, indicating that all indicators met the required standard.

In the final three-factor model, Factor 1 and Factor 3 are completely consistent with the “Hedonism” and “Eudaimonism” dimensions in the original scale, respectively. F2 is a new dimension, and the items all come from newly added items. Three items are derived from Other-hedonism, and two items are derived from Other-eudaimonism. As they all focus on others, they are tentatively named the “Otherism” dimension.

### 3.3. Test of Scale Structure

#### 3.3.1. Confirmatory Factor Analysis

A confirmatory factor analysis was conducted on the structure of the measurement tool using Sample 2. Based on theoretical assumptions and previous research findings, three hypothetical models were proposed:
Model 1: A two-factor model consisting of the Hedonism and Eudaimonism dimensions. The Hedonism dimension includes five items from the original scale and three additional items from the Other-hedonism dimension, totaling eight items. The Eudaimonism dimension includes five items from the original scale and two additional items from the Other-eudaimonism dimension, totaling seven items.Model 2: A three-factor model based on the exploratory factor analysis conducted on Sample 1, which includes the Hedonism, Eudaimonism, and Otherism dimensions, each with five items.Model 3: A four-factor model based on the findings from our previous study [11], which includes the Hedonism, Eudaimonism, Other-hedonism, and Other-eudaimonism dimensions. The Hedonism and Eudaimonism dimensions each include five items, the Other-hedonism dimension includes three items, and the Other-eudaimonism dimension includes two items.

The fitting indicators for the three models are shown in Table 2.

As shown in the above table, the two-factor model had the worst fit among all indicators and did not meet the standards, while the indicators of the three-factor and four-factor models were within an acceptable range. Moreover, all factors in both models had significant and high factor loadings (greater than 0.60, *ps* < 0.001). Therefore, further investigation was conducted on both models.

#### 3.3.2. Convergent and Discriminant Validity Tests

First, the convergent validity of the two models was tested. Convergent validity was mainly evaluated by two indicators: average variance extracted (AVE) and composite reliability (CR), both of which were calculated based on the factor loadings obtained from the confirmatory factor analysis. The AVE and CR values for both models are shown in Table 3.

Generally, if the AVE of a factor is greater than 0.50 and the CR value is greater than 0.70, it indicates that the convergent validity of the factor is high. However, as shown in Table 3, the Other-eudaimonism factor in the four-factor model did not meet both criteria.

Further investigation was conducted to evaluate the discriminant validity of the three-factor and four-factor models. Discriminant validity can be evaluated by two indicators: maximum shared squared variance (MSV) and average shared squared variance (ASV). If the MSV and ASV values of a factor are both smaller than its AVE value, it indicates that the factor has good discriminant validity. Table 3 also shows the MSV and ASV values for each factor in both models, revealing that the three-factor model had good discriminant validity, while the four-factor model did not meet the MSV and ASV criteria for all four factors and the Other-eudaimonism factor.

Another method to investigate discriminant validity is to compare the correlation between each factor and the square root of its AVE. The correlation indicators for the three-factor and four-factor models are shown in Table 4 and Table 5, respectively.

In general, if there are significant correlations between the factors in the model and the correlation coefficients are all less than the square root of their corresponding AVEs, it indicates that there is sufficient discriminant validity among the latent variables. Therefore, the model has good discriminant validity.

As shown in the above tables, the correlation coefficients among the factors in the three-factor model met the criteria, but the square root of the AVE value for the Other-eudaimonism factor in the four-factor model was 0.72, which is less than the maximum absolute value of the correlation coefficients among the factors (0.79). Thus, the discriminant validity of the Other-eudaimonism factor was inadequate.

Overall, based on the comprehensive consideration of the above indicators, the three-factor model of happiness orientation had better convergent validity and discriminant validity. Thus, in this study, the three-factor model was used for further research and the scale was named the “Hedonic, Eudaimonic, and Otheristic Motives for Activities Scale-Chinese (HEOMA-C)”.

### 3.4. Other Indicator Tests

#### 3.4.1. Reliability 

The split-half reliability and internal consistency coefficients of the scale were tested using data from Sample 1 and Sample 2. The results are shown in Table 6. The total score and each dimension of the scale performed well in terms of internal consistency and split-half reliability in both samples.

The retest reliability of the scale was tested using data from Sample 3. In the four follow-up tests, the retest reliability of the whole scale and the three dimensions (Hedonism, Eudaimonism, and Otherism) was 0.58 ~ 0.78, 0.63 ~ 0.80, 0.60 ~ 0.79, and 0.56 ~ 0.76.

#### 3.4.2. Criterion Validity 

The criterion validity of the scale was examined using data from Sample 2 and Sample 4. We used the OHS (*N* = 511), the BAWB scale (*N* = 200), and the CoDI scale (*N* = 200) as criterion measures. The luck dimension of the CoDI scale had a weak association with HEOMA-C, so it was not included in the assessment. The results showed significant positive correlations between the HEOMA-C and corresponding dimensions of the OHS, the BAWB scale, and the CoDI scale, including Eudaimonism with meaning and engagement of the OHS, self-development and contribution to others of the BAWB scale, and personal development and purpose of the CoDI scale; Hedonism with pleasure of the OHS, pleasure and avoidance of negative of the BAWB scale, and positive attitude, joy and desires and absence of negativity of the CoDI scale; and Otherism with meaning and engagement of the OHS, contribution to others of the BAWB scale, and belonging of the CoDI scale. As shown in Table 7, the results indicated that the HEOMA-C has good criterion validity.

#### 3.4.3. Longitudinal Measurement Invariance

The longitudinal measurement invariance was tested by Mplus using data from Sample 3. The configural invariance model (M1), metric invariance model (M2), and scalar invariance model (M3) were constructed and verified. According to the standard proposed by Chen [21], when the metric invariance model is compared with the configural invariance model and the scalar invariance model, if ΔCFI < 0.01 and ΔRMSEA < 0.02, then it means that they are invariant. The results showed that the HEOMA-C had good longitudinal measurement invariance in the four tests (see Table 8).

## 4. Discussion

In the present study, we revised and validated the HEOMA-C based on the OTH structure in Chinese culture [11]. After EFA and CFA, we found that the three-factor model was more reasonable. We named the three dimensions Hedonism, Eudaimonism, and Otherism and defined Otherism as “the pursuit of the harmony of the group and achieving happiness by fulfilling their responsibilities in the group”. For individuals who score high on this dimension, regardless of whether their specific behavior is to pursue common pleasure with others or create value for others, the core of their real concern is relationships. The HEOMA-C had good reliability and validity.

The present study took Otherism as a new dimension of OTH, which coincided with previous studies. Previous researchers have mentioned the importance of social relationships to happiness [22,23], explored the influence of intimate others and social support on OTH [15,24,25], and found different connotations of well-being (both hedonic and eudaimonic) under individualistic culture and collectivist culture. Oyserman et al. found through meta-analysis that in an individualistic culture, people regarded achieving personal goals and having a greater sense of control as the core of happiness [10]. In a collectivist culture, people saw fulfilling their obligations and duties as the key to happiness [26,27,28]. This shows that the pursuit of one’s own success and the pursuit of the value one brings to others are two different pursuits.

Several limitations should also be recognized. First, the participants in this study were all young and middle-aged (18–52 years old), providing evidence for the OTH throughout the age period; however, the results should be applied to other populations with caution. In the future, large sample surveys of all age groups can be further carried out to investigate the distribution of OTH among Chinese adults. Second, we only revised the measurement tool but did not explore which factors affected the OTH. Future research can explore the influencing factors of OTH, the relationship between OTH and well-being, and effective interventions to enhance OTH.

Despite these limitations, this study provides important theoretical and practical implications. We revised the HEOMA-C, providing an effective measurement tool for follow-up research on OTH in Chinese culture. The result showed that “relationship” should be included in the study of well-being in the context of collectivist cultures. This study also illustrates the importance of caring about others. Spending time with others, creating value for others, making others happy, and struggling with others can also bring us happiness in our lives.

## 5. Conclusions

The present study revised and validated the Hedonic, Eudaimonic, and Otheristic Motives for Activities Scale-Chinese (HEOMA-C) in Chinese culture. The results showed that in the context of Chinese culture, the OTH has three dimensions: Hedonism, Eudaimonism, and Otherism. Among them, Otherism is a new dimension, which means “the pursuit of the harmony of the group and achieving happiness by fulfilling their responsibilities in the group”.

## Figures and Tables

**Table 1 behavsci-13-00746-t001:** Factor loading matrix (*N* = 479).

IN	F1	F2	F3	COM	DI	TIC
1	0.80	0.12	0.14	0.68	0.43	0.82 ***
3	0.80	0.01	0.25	0.70	0.39	0.85 ***
4	0.85	0.10	0.07	0.73	0.47	0.79 ***
7	0.71	0.23	0.23	0.61	0.41	0.82 ***
9	0.78	0.11	0.21	0.67	0.40	0.82 ***
11	0.19	0.72	0.13	0.56	0.42	0.76 ***
12	0.16	0.85	0.08	0.75	0.47	0.84 ***
13	0.22	0.84	0.14	0.77	0.42	0.86 ***
14	0.02	0.62	0.20	0.43	0.43	0.70 ***
15	−0.03	0.76	0.17	0.61	0.37	0.76 ***
2	0.31	0.08	0.71	0.60	0.33	0.76 ***
3	0.18	0.12	0.72	0.56	0.35	0.75 ***
5	0.14	0.12	0.80	0.68	0.40	0.81 ***
8	0.10	0.31	0.65	0.53	0.35	0.73 ***
10	0.16	0.19	0.79	0.68	0.39	0.81 ***
EV	3.41	3.16	2.98			
CPVE (%)	22.72	43.80	63.78			

Note: *** *p* < 0.001. F1 = Factor 1, F2 = Factor 2, F3 = Factor 3, COM = commonality, DI = discrimination, TIC = total test correlation, EV = eigen value, CPVE = cumulative percentage of explained variance.

**Table 2 behavsci-13-00746-t002:** Fit indices for the confirmatory factor analysis validation mode (*N* = 511).

Model	χ^2^	*df*	χ^2^/*df*	RMSEA	SRMR	CFI	TLI
two-factor	1400.211	89	15.73	0.17	0.13	0.71	0.66
three-factor	389.025	86	4.52	0.08	0.06	0.93	0.92
four-factor	372.500	83	4.49	0.08	0.06	0.94	0.92

Note: CFI = comparative fit index, TLI = Tucker–Lewis index, SRMR = standardized root mean square residual, RMSEA = root mean square error of approximation.

**Table 3 behavsci-13-00746-t003:** Convergent and discriminant validity scores of two models (*N* = 511).

	Factor	AVE	CR	MSV	ASV
three-factor model	Hedonism	0.64	0.90	0.31	0.52
	Eudaimonism	0.55	0.86	0.26	0.50
	Otherism	0.61	0.88	0.31	0.53
four-factor model	Hedonism	0.64	0.90	1.03	0.62
	Eudaimonism	0.55	0.86	1.03	0.62
	Other-hedonism	0.66	0.85	1.03	0.69
	Other-eudaimonism	0.44	0.61	1.03	0.71

Note: AVE = average variance extracted, CR = composite reliability, MSV = maximum of shared squared variance, ASV = average of shared squared variance.

**Table 4 behavsci-13-00746-t004:** Correlations and AVE square roots for the three-factor model (*N* = 511).

	Hedonism	Eudaimonism	Otherism
Hedonism	0.80		
Eudaimonism	0.43 ***	0.74	
Otherism	0.51 ***	0.50 ***	0.78

Note: *** *p* < 0.001.

**Table 5 behavsci-13-00746-t005:** Correlations and AVE square roots for the four-factor model (*N* = 511).

	Hedonism	Eudaimonism	Other- Hedonism	Other-Eudaimonism
Hedonism	0.799			
Eudaimonism	0.433 ***	0.742		
Other-hedonism	0.529 ***	0.484 ***	0.808	
Other-eudaimonism	0.418 ***	0.454 ***	0.793 ***	0.724

Note: *** *p* < 0.001.

**Table 6 behavsci-13-00746-t006:** Reliability indices of the scale.

Item	ICC		SHR	
	Sample 1 (*N* = 479)	Sample 2 (*N* = 511)	Sample 1 (*N* = 479)	Sample 2 (*N* = 511)
Hedonism	0.88	0.90	0.88	0.91
Eudaimonism	0.83	0.86	0.79	0.82
Otherism	0.84	0.87	0.80	0.84
Total score	0.88	0.91	0.73	0.84

Note: ICC = internal consistency coefficient, SHR = split-half reliability.

**Table 7 behavsci-13-00746-t007:** The criterion validity of HEOMA-C.

Dimension	*M*	*SD*	α	Eudaimonism	Hedonism	Otherism
Orientation to Happiness Scale						
Meaning	3.48	0.95	0.82	0.64 ***	0.04	0.22 **
Pleasure	3.54	0.97	0.79	−0.11	0.45 ***	0.07
Engagement	3.14	0.97	0.80	0.037 ***	0.02	0.15 *
Beliefs about Well-Being						
Pleasure	5.24	1.20	0.78	−0.02	0.60 ***	0.20
Avoidance of negative	4.65	1.41	0.81	−0.01	0.53 ***	0.19
Self-development	5.88	1.00	0.70	0.67 ***	−0.01	0.23 *
Contribution to others	5.55	1.11	0.80	0.46 ***	−0.16	0.37 ***
HWB Definitions						
Positive attitude	5.16	0.83	0.61	0.18	0.61 ***	0.15
Tranquility	4.82	0.82	0.59	0.28 **	0.12	0.21 *
Joy and desires	4.73	1.02	0.68	0.09	0.60 ***	0.25 *
Absence of negativity	4.84	1.24	0.79	0.06	0.59 ***	0.40 ***
Personal development	4.91	1.05	0.65	0.58 ***	0.03	0.19
Purpose	5.16	0.78	0.77	0.53 ***	0.12	0.15
Belonging	4.83	1.05	0.74	0.24 *	0.08	0.57 ***
HWB Intentions						
Positive attitude	5.05	0.79	0.68	0.24 *	0.55 ***	0.18
Tranquility	4.75	0.80	0.51	0.29 **	0.18	0.08
Joy and desires	4.58	1.25	0.67	0.14	0.60 ***	0.28 **
Absence of negativity	4.22	1.36	0.85	0.11	0.45 **	0.29 **
Personal development	4.95	0.91	0.77	0.55 ***	0.06	0.16
Purpose	5.13	0.83	0.70	0.45 ***	0.18	0.18
Belonging	4.80	1.04	0.83	0.25 **	0.13	0.53 ***

Note: * *p* < 0.05, ** *p* < 0.01, *** *p* < 0.001, HWB = happiness and well-being.

**Table 8 behavsci-13-00746-t008:** The longitudinal measurement invariance of HEOMA-C *(N* = 363).

	Model Fit Index					Comparative Fit Index				
	χ^2^	*df*	CFI	TLI	RMSEA	Δχ^2^ (Δ*df*)	ΔCFI	ΔTLI	ΔRMSEA	Invariant
Eudaimonism										
M1	29.06	16	0.997	0.992	0.043	-	-	-	-	-
M2	39.48	28	0.997	0.996	0.030	10.42 (12)	0.000	0.004	−0.007	yes
M3	59.52	40	0.995	0.995	0.033	20.04 (12)	−0.002	0.001	0.003	yes
Hedonism										
M1	25.72	10	0.997	0.988	0.060	-	-	-	-	-
M2	67.45	22	0.991	0.984	0.068	41.73 (12)	−0.006	−0.004	0.008	yes
M3	128.24	34	0.982	0.979	0.072	60.79 (12)	−0.009	−0.005	0.004	yes
Otherism										
M1	27.00	14	0.997	0.992	0.049	-	-	-	-	-
M2	66.56	26	0.992	0.987	0.057	39.56 (12)	−0.005	−0.005	0.008	yes
M3	116.58	38	0.984	0.983	0.066	50.02 (12)	−0.008	−0.004	0.009	yes

Note: CFI = comparative fit index, TLI = Tucker–Lewis index, RMSEA = root mean square error of approximation.

## Data Availability

The data presented in this study are available on request from the corresponding author.

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
