# Peer review of "A New Factor “Otherism” Added to the Hedonic and Eudaimonic Motives for Activities Scale (HEMA) in Chinese Culture"

_behavsci, 2023, doi:10.3390/bs13090746_

Round 1

Reviewer 1 Report

Very enjoyable manuscript in general. I suggest to further develop the consequences of your overall findings to mainstream cross cultural research, such as how Otherism could influence outcome variables in other lines of research.

Author Response

- Very enjoyable manuscript in general. I suggest to further develop the consequences of your overall findings to mainstream cross cultural research, such as how Otherism could influence outcome variables in other lines of research.

Response: Thank you for your careful reading and encouragement. We appreciate your comments and have revised the manuscript accordingly.

(p. 13) The major strengths of the study are that we first explored the structure of the OTH of Chinese adults and discovered the dimension Otherism, which is a certain theoretical innovation. The results of this study suggest that relationship-related dimension should be included in the study of well-being under collectivist culture, and the influence of otherism on well-being and other related outcome variables should also be paid attention to in cross-cultural research (Gaston-Breton et al., 2021; Hitokoto & Uchida, 2015; Krys et al., 2019).

Reviewer 2 Report

The manuscript presents an interesting investigation of paths to happiness in collectivist cultures by (1) using a qualitative study approach and (2) developing and validating an extended Chinese version of the HEMA-Scale. I appreciate the general aim to investigate paths to happiness in collectivist cultures and also consider the novel factor “Otherism” as useful. Moreover, the scale development and translation process in Study 2 adhered to state-of-the-art methodologies. Although the study generally has potential to contribute to the happiness literature, I have several suggestions on how to improve the clarity and rigor of the manuscript.

Major Issues:

-       The introduction needs improvement in clarity, particularly in defining different happiness conceptions. The distinction between hedonia and eudaimonia should be more clearly articulated. For example, to which of the concepts described do Ryan and Deci's basic psychological needs refer to? The notion that some researchers regard Hedonia "as a way of life" and Eudaimonia "as a way of behaving" is rather vague. Additionally, the sentence about interventions targeting motivations lacks context and needs clarification.

-       When mentioning previous studies on OTH in China, the paper should outline the important cultural differences found in those studies. 

-       The method description for Study 1 lacks transparency and leaves several questions unanswered. For example, I do not understand the meaning of phrases in the section “Instumentation” like “understanding of happiness and life pursuit” or “macroscopic descriptions and specific events”. It might be helpful to provide the specific wording for the instruction.

-       Further, combining different aspects described in the "Instrumentations" section in the qualitative analysis may be problematic, as they seem to capture distinct aspects (i.e., how people describe happiness and in which specific behaviors they engage to pursue happiness is not the same).

-       The duration to complete the semi-structured interview for each participant should be reported.

-       The coding procedure needs to be described more thoroughly and transparently. Ideally, references to the precise method used for coding should be provided. The paper should address whether the coding was repeated by others to determine intercoder agreement, as this is critical for establishing reliability of the coding process.

-       The results description for Study 1 lacks clarity and coherence. The authors should explain why only 51 out of 421 fragments were named and why only eight concepts were formed. The process of forming categories and the criteria used to distinguish between different concepts should be clearly outlined. How exactly were categories formed and according to what criteria were concepts categorized as different from one another? For example, how did the coders come to the conclusion that "Comfort and relaxation" must be distinguished from "Pleasure and satisfaction"? I also don't understand the distinction between "Shared Pleasure" and "Good Relationships" at all (the description “Shared pleasure refers to the pursuit of pleasure and comfort in interacting with significant others, and Good relationships refers to the pur- suit of relationships that make one feel pleasant and comfortable.” is not very helpful).

-       Study 2: The OHS by Peterson et al. chosen for validation purposes has severe limitations (see for example Henderson et al., 2014). Psychometric problems aside, this scale does not contain any factor that could be related to "Otherism" (as also shown by the rather low correlations between this factor and the OTH dimensions reported in the manuscript). Probably this scale was chosen because it is the only relevant scale available in Chinese? However, I think it would be important to  (additionally) assess other scales that measure different paths to happiness, e.g., the BAWB scale (McMahan & Estes, 2011) or the CoDI scale (Krasko et al., 2022). Both scales also include paths to happiness that appear to be related to "Otherism."

-       The manuscript mentions retesting (p. 7) but fails to provide any further information or report on the retest reliability.

Minor Issues

-       p. 7: “To test the psychometric properties of the scale we developed, we collected another sample.” ïƒ  ‘we developed’ seems to be redundant here.

-       P. 10: “A total of 539 valid questionnaires were collected by T1,…” what does ‘valid’ mean in this context?

-       Correct the spelling of "Eudaimonia" in Table 8.

Referenes

Henderson, L. W., Knight, T., & Richardson, B. (2014). The hedonic and eudaimonic validity of the Orientations to Happiness Scale. Social Indicators Research115(3), 1087–1099. https://doi.org/10.1007/s11205-013-0264-4

Krasko, J., Intellisano, S., & Luhmann, M. (2022). When happiness is both joy and purpose: The complexity of the pursuit of happiness and well-being is related to actual well-being. Journal of Happiness Studies23, 3233–3261. https://doi.org/10.1007/s10902-022-00541-2

McMahan, E. A., & Estes, D. (2011). Measuring lay conceptions of well-being: The beliefs about well-being scale. Journal of Happiness Studies12(2), 267–287. https://doi.org/10.1007/s10902-010-9194-x

Author Response

Dear Reviewer 2,

Thank you very much for giving us the opportunity to revise our manuscript entitled “Pursuing harmony and fulfilling responsibility: A new factor added to the Hedonic and Eudaimonic Motives for Activities Scale (HEMA) in Chinese culture” (Manuscript Number: behavsci-2502894). We greatly appreciate your comments and suggestions. Your expertise was very helpful for us in improving the quality of the paper. We have revised the manuscript and would like to resubmit it for publication in the Behavior Science.

We hope that the revised manuscript addresses most of the concerns that you raised and incorporates most of your suggestions. As a result of your valuable suggestions, the quality of the manuscript has been significantly improved. We hope to receive a favorable response from you. Please find our point-by-point responses attached.

Responses to Reviewer 2

The manuscript presents an interesting investigation of paths to happiness in collectivist cultures by (1) using a qualitative study approach and (2) developing and validating an extended Chinese version of the HEMA-Scale. I appreciate the general aim to investigate paths to happiness in collectivist cultures and also consider the novel factor “Otherism” as useful. Moreover, the scale development and translation process in Study 2 adhered to state-of-the-art methodologies. Although the study generally has potential to contribute to the happiness literature, I have several suggestions on how to improve the clarity and rigor of the manuscript.

Response: Thank you for your careful reading and encouragement. We appreciate your comments and have revised the manuscript accordingly. Please see the following point-by-point responses and the revised manuscript.

Major Issues:

- The introduction needs improvement in clarity, particularly in defining different happiness conceptions. The distinction between hedonia and eudaimonia should be more clearly articulated.

Response: Thank you for your valuable advice. We have made specific revisions based on your suggestions as follows.

-1) For example, to which of the concepts described do Ryan and Deci's basic psychological needs refer to?

Response: We have provided explanations for autonomy, competence, and relatedness respectively.

(p. 1) Ryan and Deci (2000) also proposed that happiness is the satisfaction of the needs of autonomy (referring to the degree of independence and self-determination in one's behavior), competence (referring to an individual's perception and confidence in their own abilities) and relatedness (referring to the relationships and connections an individual has with others).

-2) The notion that some researchers regard Hedonia "as a way of life" and Eudaimonia "as a way of behaving" is rather vague.

Response: We have made a more specific explanation. Hedonia is commonly defined using emotions and feelings as indicators, while eudaimonia is commonly defined using behaviors as indicators.

(p. 2) For example, hedonic measurement indicators are defined in terms of emotions and feelings (e.g., life satisfaction), which considered as an outcome variable. While eudaimonic measurement indicators are defined in terms of behavior (e.g., striving for the top; Vittersø, 2003, 2004), which considered as a predictive variable.

-3) Additionally, the sentence about interventions targeting motivations lacks context and needs clarification.

Response: Thank you for pointing out this issue. After careful consideration, we have decided to delete this sentence, as it does not affect the content of this paragraph or the overall article, and the coherence between the surrounding context is also enhanced.

- When mentioning previous studies on OTH in China, the paper should outline the important cultural differences found in those studies.

Response: Thank you for your valuable advice. We first elucidated the differences between individualistic and collectivist cultures, and then, by incorporating cultural variations, further explained the core elements of happiness as perceived by individuals in different cultural environments.

(p. 2) The findings showed that in an individualistic cultural environment, people put more emphasis on individual freedom and independence. People believe that achieving personal goals and having more control over their lives are the core elements of happiness. However, in a collectivist cultural environment, people pay more attention to the relationship and responsibility between the individual and the social group. People believe that contributing to the collective and fulfilling individual social obligations is the key to happiness, and value spiritual enrichment rather than hedonic satisfaction.

- The method description for Study 1 lacks transparency and leaves several questions unanswered. For example, I do not understand the meaning of phrases in the section “Instumentation” like “understanding of happiness and life pursuit” or “macroscopic descriptions and specific events”. It might be helpful to provide the specific wording for the instruction.

- Further, combining different aspects described in the "Instrumentations" section in the qualitative analysis may be problematic, as they seem to capture distinct aspects (i.e., how people describe happiness and in which specific behaviors they engage to pursue happiness is not the same).

Response: Thank you for your valuable advice. We clarified this sentence.

(p. 3) A semi-structured interview outline was developed and revised through pilot testing and external review prior to data collection. The interview outline was divided into three parts: (1) basic personal information, including age, gender, and marital status; (2) understanding of the happiness they pursue, describing the ideal life scenario and general state of happiness; and (3) specific actions in the pursuit of happiness, describing the specific behaviors they will take to achieve happiness described above.

- The duration to complete the semi-structured interview for each participant should be reported.

Response: Thank you for pointing out this issue. We added the duration.

(p. 3) The average duration of the interviews was 14.67 minutes (SD = 2.69), ranging from 10.83 to 18.48 minutes.

- The coding procedure needs to be described more thoroughly and transparently. Ideally, references to the precise method used for coding should be provided. The paper should address whether the coding was repeated by others to determine intercoder agreement, as this is critical for establishing reliability of the coding process.

Response: Thank you for your valuable advice. We added the reference and explained the procedure of intercoder agreement.

(p. 3) The analysis process derived from the coding and conceptualization processes of Grounded Theory (GT) (Glaser & Strauss, 2019). First, two researchers independently coded transcripts line-by-line using an open-coding technique, merged fragments with the same meaning, and classified and summarized the fragments to form the concepts. For example, "Sometimes I want to relax and don't think about anything." was coded as the fragment "relax". Together with similar fragments like "sleep in" and "leisure reading", it is classified to form the concept Comfort & Relaxation. After this, researchers independently raised the agreed-upon concepts into axial codes by categorizing and naming the concepts with close semantic connections and forming the main categories. For example, Comfort & Relaxation and Pleasure & Satisfaction formed the main category Self-hedonism. In the final step, selective coding, the team connected multiple categories into core categories. For example, Self-hedonism and Self-eudaimonism formed the core category Self-focused. In the analysis process, we carefully considered the balance of various voices. Differences were resolved through discussion and mutual consensus was obtained.

- The results description for Study 1 lacks clarity and coherence. The authors should explain why only 51 out of 421 fragments were named and why only eight concepts were formed.

Response: Thank you for your careful reading. We clarified this sentence to better explain it.

(p. 3) Through line-by-line coding, a total of 421 fragments were obtained. Researchers merged fragments with the same meaning and ended up with 51 fragments. Then, the fragments were classified and summarized to form the following eight concepts.

- The process of forming categories and the criteria used to distinguish between different concepts should be clearly outlined. How exactly were categories formed and according to what criteria were concepts categorized as different from one another?

For example, how did the coders come to the conclusion that "Comfort and relaxation" must be distinguished from "Pleasure and satisfaction"? I also don't understand the distinction between "Shared Pleasure" and "Good Relationships" at all (the description “Shared pleasure refers to the pursuit of pleasure and comfort in interacting with significant others, and Good relationships refers to the pursuit of relationships that make one feel pleasant and comfortable.” is not very helpful)

Response: Thank you for your valuable advice. We modified these statements to better explain these concepts.

(p. 4) Comfort & Relaxation and Pleasure & Satisfaction. The former means one's pursuit of a stress-free state, as a participant said, "Sometimes I want to completely relax and don't think about anything." The latter means one's pursuit of fun and pleasure, as one participant said, "I like watching funny shows and just giggling."

(p. 4) Shared pleasure and Good relationships. Shared pleasure refers to the pursuit of pleasure and comfort in interacting with significant others, as some participants said, "Playing with a group of friends makes me happy." "I like to be with my boyfriend. I feel very relaxed even if I just watch him play games." Good relationships refers to the pursuit of high-quality relationships, with an emphasis on the relationship itself. These relation-ships include family, friends, lovers, etc., as some participants said, "I wish I could spend more time with my family." "I like to make new friends."

- Study 2: The OHS by Peterson et al. chosen for validation purposes has severe limitations (see for example Henderson et al., 2014). Psychometric problems aside, this scale does not contain any factor that could be related to "Otherism" (as also shown by the rather low correlations between this factor and the OTH dimensions reported in the manuscript). Probably this scale was chosen because it is the only relevant scale available in Chinese? However, I think it would be important to (additionally) assess other scales that measure different paths to happiness, e.g., the BAWB scale (McMahan & Estes, 2011) or the CoDI scale (Krasko et al., 2022). Both scales also include paths to happiness that appear to be related to "Otherism."

Response: Thank you for your valuable advice. After careful consideration, we decided to change the criterion. We collected another sample and used the BAWB scale and the CoDI scale to examine criterion validity.

(p. 11) Criterion validity. The criterion validity of the scale was examined using data from Sample 4. We used BAWB scale (McMahan & Estes, 2011) and the CoDI scale (Krasko et al., 2022) as criterion measure. The Luck dimension of the CoDI scale has weak association with HEOMA-C, so it was not included in the assessment. The results showed significant positive correlations between the HEOMA-C and corresponding dimensions of BAWB scale and the CoDI scale, including Eudaimonism with Self-development and Contribution to others of the BAWB scale and Personal development and Purpose of the CoDI scale, Hedonism with Pleasure and Avoidance of negative of the BAWB scale and Positive attitude, Joy and desires, Absence of negativity of the CoDI scale, as well as Otherism with Contribution to others of the BAWB scale and Belonging of the CoDI scale. As shown in table 8, the results indicated that the HEOMA-C has good criterion validity.

.

Table 8. The criterion validity of HEOMA-C (N =200).

Dimension

M

SD

α

Eudaimonism

Hedonism

Otherism

Beliefs about Well-Being

Pleasure

5.24

1.20

.78

-.02

.60***

.20

Avoidance of negative

4.65

1.41

.81

-.01

.53***

.19

Self-development

5.88

1.00

.70

.67***

-.01

.23*

Contribution to others

5.55

1.11

.80

.46***

-.16

.37***

HWB Definitions

Positive attitude

6.13

0.83

.61

.18

.61***

.15

Tranquility

5.81

0.82

.59

.28**

.12

.21*

Joy and desires

5.65

1.02

.68

.09

.60***

.25*

Absence of negativity

5.37

1.24

.79

.06

.59***

.40***

Personal development

5.85

1.05

.65

.58***

.03

.19

Purpose

6.15

0.78

.77

.53***

.12

.15

Belonging

5.77

1.05

.74

.24*

.08

.57***

HWB Intentions

Positive attitude

6.04

0.79

.68

.24*

.55***

.18

Tranquility

5.73

0.80

.51

.29**

.18

.08

Joy and desires

5.44

1.25

.67

.14

.60***

.28**

Absence of negativity

5.04

1.36

.85

.11

.45**

.29**

Personal development

5.97

0.91

.77

.55***

.06

.16

Purpose

6.12

0.83

.70

.45***

.18

.18

Belonging

5.73

1.04

.83

.25**

.13

.53***

- The manuscript mentions retesting (p. 7) but fails to provide any further information or report on the retest reliability.

Response: Thank you for your careful reading. We clarified this sentence to better explain it.

(p. 11) The retest reliability of the scale was tested using data from Sample 3. In the four follow-up tests, the retest reliability of the whole scale and the three dimensions (Hedonism, Eudaimonism, Otherism) was 0.58 ~ 0.74, 0.63 ~ 0.80, 0.60 ~ 0.79, 0.56 ~ 0.76.

Minor Issues:

- p. 7: “To test the psychometric properties of the scale we developed, we collected another sample.” à ‘we developed’ seems to be redundant here.

Response: Thank you for pointing out this issue. We deleted 'we developed'.

(p. 7) To test the retest reliability and the longitudinal measurement invariance, we collected another sample.

- P. 10: “A total of 539 valid questionnaires were collected by T1,…” what does ‘valid’ mean in this context?

Response: Thank you for your careful reading. This means that the participant completed all the questions in the questionnaire. The excluded data due to incomplete or inaccurate responses. We added the explanation.

(p. 6) A total of 539 valid questionnaires were collected by T1, including 199 males (36.9%) and 340 females (63.1%), with a mean age of 24.98 years old (SD = 6.62), ranging from 18 to 52 years old. The excluded data due to incomplete or inaccurate responses.

- Correct the spelling of "Eudaimonia" in Table 8.

Response: Thank you for careful reading. We corrected the spelling in Table 8.

Reference:

Henderson, L. W., Knight, T., & Richardson, B. (2014). The hedonic and eudaimonic validity of the Orientations to Happiness Scale. Social Indicators Research115(3), 1087–1099. https://doi.org/10.1007/s11205-013-0264-4

Krasko, J., Intellisano, S., & Luhmann, M. (2022). When happiness is both joy and purpose: The complexity of the pursuit of happiness and well-being is related to actual well-being. Journal of Happiness Studies23, 3233–3261. https://doi.org/10.1007/s10902-022-00541-2

McMahan, E. A., & Estes, D. (2011). Measuring lay conceptions of well-being: The beliefs about well-being scale. Journal of Happiness Studies12(2), 267–287. https://doi.org/10.1007/s10902-010-9194-x

Response: Thank you so much for sharing the references. We added them into the manuscript.

(p. 6)

The Beliefs About Well-Being Scale (BWBS)

The BWBS was developed by McMahan and Este (2010). It consists of 16 items di-vided into four dimensions, with each dimension containing four items. The dimensions are the Experience of Pleasure (e.g., Experiencing Euphoria and Pleasure), Avoidance of Negative Experience (e.g., A Lack of Painful Experiences), Self-Development (e.g., Working to Achieve One's True Potential), and Contribution to Others (e.g., Living in Ways That Benefit Others). Participants rate each item on a 7-point scale based on their own beliefs (1 = strongly disagree, 7 = strongly agree). A team of four researchers translated it into Chinese. In this study, BWBS had good internal consistency (α= 0.76).

Complexity of Happiness Definitions and Intentions (CoDI)

The CoDI was developed by Krasko, Intelisano, and Luhmann (2022). It consists of 24 items divided into eight subscales, with each subscale containing three items. Each item includes two questions: Hwb definitions (e.g., for me personally, happiness means...) and Hwb intentions (e.g., in daily life, I try...). The subscales include Absence of Negativity (e.g., not to worry), Positive Attitude (e.g., to go through life cheerfully), Tranquility (e.g., a peaceful state of mind), Personal Development (e.g., to develop be-yond oneself), Luck (e.g., to be favored by luck or fate), Joy and Desires (e.g., experiencing a great deal of pleasure), Purpose (e.g., to have a clear meaning in life), and Be-longing (e.g., to feel close to others). Participants rate each item on a 6-point scale based on their own beliefs (1 = does not apply, 6 = applies completely). A team of four re-searchers translated it into Chinese. In this study, CoDI had good internal consistency (α definition = 0.75, α intention = 0.81).

Round 2

Reviewer 2 Report

Overall, my comments were implemented. I have a few additional comments.

-       My previous comment: -1) For example, to which of the concepts described do Ryan and Deci's basic psychological needs refer to?; Response: We have provided explanations for autonomy, competence, and relatedness respectively.; New comment: What I was actually referring to with my comment is that it should be clearer that basic psychological needs refer to an eudaimonic perspective.

-       Author response: After careful consideration, we decided to change the criterion. We collected another sample and used the BAWB scale and the CoDI scale to examine criterion validity.; New comment: I did not mean that the OHS had any value at all and therefore I would keep it as criterion for the validity analyses. After all, despite its weaknesses, it is one of the most commonly used scales in the research area and thus allows for comparison with the results of other studies. The additional inclusion of the BAWB and Codi scales clearly demonstrate the validity of the HEOMA-C more convincingly, especially for the "Otherism" factor.

-       In several places a space is missing before or after the references, e.g., “Some researchers have tested the relevant indicators of the HEMA scale[19]…”. 

-       P. 6: A “s” is missing in “Estes”: “The BWBS was developed by McMahan and EsteS[31].”

-       Please introduce the abbreviation “Hwb” (or just do not use it)

-       Table 8: Since the Codi scale response options range from 1 to 6, I wonder how mean values above 6 come about. (e.g., for the factor “positive attitude”). This should be checked.

Author Response

Dear Reviewer 2,

Thank you very much for giving us the opportunity to revise our manuscript entitled “Pursuing harmony and fulfilling responsibility: A new factor added to the Hedonic and Eudaimonic Motives for Activities Scale (HEMA) in Chinese culture” (Manuscript Number: behavsci-2502894). We greatly appreciate your comments and suggestions. Your expertise was very helpful for us in improving the quality of the paper. We have revised the manuscript and would like to resubmit it for publication in the Behavioral Sciences.

We hope that the revised manuscript addresses most of the concerns that you raised and incorporates most of your suggestions. As a result of your valuable suggestions, the quality of the manuscript has been significantly improved. We hope to receive a favorable response from you.

Please find our point-by-point responses attached.
